



# Automated avalanche mapping from SPOT 6/7 satellite imagery: results, evaluation, potential and limitations

Elisabeth D. Hafner[1,2,3], Patrick Barton[3], Rodrigo Caye Daudt[3], Jan Dirk Wegner[3,4], Konrad Schindler[3], and Yves Bühler[1,2]

[1]WSL Institute for Snow and Avalanche Research SLF, Davos Dorf, 7260, Switzerland
[2]Climate Change, Extremes, and Natural Hazards in Alpine Regions Research Center CERC, Davos Dorf, 7260, Switzerland
[3]EcoVision Lab, Photogrammetry and Remote Sensing, ETH Zurich, Zurich, 8092, Switzerland
[4]Institute for Computational Science, University of Zurich, Zurich, 8057, Switzerland

**Correspondence:** Elisabeth D. Hafner (elisabeth.hafner@slf.ch)

**Abstract.** Spatially dense and continuous information on avalanche occurrences is crucial for numerous safety related applications such as avalanche warning, hazard zoning, hazard mitigation measures, forestry, risk management and numerical simulations. This information is today still collected in a non-systematic way by observers in the field. Current research has explored and proposed applying remote sensing technology to fill this information gap by providing spatially continuous in-
formation on avalanche occurrences over large regions. Previous investigations have confirmed the high potential of avalanche mapping from remote sensed imagery to complement existing databases. Currently, the bottleneck for fast data provision from optical data is the time-consuming manual mapping. In our study we deploy a slightly adapted DeepLabV3+, a state-of-the-art deep learning model, to automatically identify and map avalanches in SPOT6/7 imagery from 24 January 2018 and 16 January 2019. We relied on 24'778 manually annotated avalanche polygons split into geographically disjoint regions for training, vali-
dating and testing. Additionally, we investigate generalization ability by testing our best model configuration on SPOT 6/7 data from 6 January 2018 and comparing to avalanches we manually annotated for that purpose. To assess the quality of the model results, we investigate the probability of detection (POD), the positive predictive value (PPV) and the F1-score. Additionally, we assessed the reproducibility of manually annotated avalanches in a small subset of our data. We achieved an average POD of 0.610, PPV of 0.668 and an F1-score of 0.625 in our test areas and found an F1-score in the same range for avalanche outlines
annotated by different experts. Our model and approach are an important step towards a fast and comprehensive documentation of avalanche periods from optical satellite imagery in the future, complementing existing avalanche databases. This will have a large impact on safety related applications, making mountain regions safer.

## 1 Introduction

Information about occurred avalanches, their location and dimensions are pivotal for many applications such as avalanche
warning, hazard zoning, hazard mitigation infrastructure, forestry, risk management and numerical simulations (e.g. Meister, 1994; Rudolf-Miklau et al., 2015; Bebi et al., 2009; Bründl and Margreth, 2015; Christen et al., 2010; Bühler et al., 2022). Currently this information is reported and collected unsystematically by observers and (local) avalanche warning services. In



recent years different groups have proposed to use remote sensing to fill that gap and provide spatially continuous, complete maps of avalanche occurrences over some region of interest (Bühler et al., 2009; Lato et al., 2012; Eckerstorfer et al., 2016; Korzeniowska et al., 2017).

It has been shown that avalanches can be identified with sufficient reliably from optical data (e.g., Bühler et al., 2019) or Synthetic Aperture Radar (SAR; e.g., Eckerstorfer et al., 2016; Abermann et al., 2019), with varying degrees of completeness depending on the sensor and the size of the avalanches (Hafner et al., 2021). To bypass the time-consuming manual mapping, several groups have explored (semi-) automatic mapping approaches. Bühler et al. (2009) used a processing chain that relies on directional, textural and spectral information to automatically detect avalanches in airborne optical data. Lato et al. (2012) and

Korzeniowska et al. (2017) applied object-based classification techniques to optical high spatial resolution data (0.25 - 0.5 m). Wesselink et al. (2017) and Eckerstorfer et al. (2019) have introduced and consequently refined an algorithm to automatically detect avalanches in Sentinel-1 SAR imagery, via changes of the backscatter between pre- and post-event images. Karbou et al. (2018) also utilized changes in backscatter to identify avalanche debris. For avalanche detection in Radarsat-2 imagery, Hamar et al. (2016) used supervised classification with a random forest classifier. On the contrary, the avalanche mapping from

optical satellite data has so far been exclusively done manually (Bühler et al., 2019; Hafner et al., 2021; Abermann et al., 2019).

The deployment of machine learning for remote sensing image analysis has seen a surge in the last decade (Ma et al., 2019). Modern deep learning methods often outperform competing ones in complex image understanding tasks, and have been used for

example to detect rock glaciers (Robson et al., 2020), landslides (Prakash et al., 2021) and crop types in fields (Cai et al., 2018). For avalanches, the use of deep learning has so far focused on Sentinel-1 imagery: Waldeland et al. (2018) applied a pre-trained ResNet (He et al., 2016) for avalanche identification by change detection using manual reference annotations. Bianchi et al. (2021) segmented avalanches with a fully convolutional U-Net (Ronneberger et al., 2015), also relying on manual annotations for training the network. Sinha et al. (2019a) proposed a fully convolutional VGG16 network (Simonyan and Zisserman, 2015)

that was trained on, and compared against, an inventory of avalanche field observations. With the same inventory, Sinha et al. (2019b) also, alternatively used a Variational Autoencoder (Kingma and Welling, 2019) for avalanche detection.

In contrast to previous studies, our work is the first to attempt to use deep learning for the detection of avalanches in *optical* satellite data. This is of major importance, as the largest avalanche mapping from remotely sensed imagery to date, with 24'778 single avalanche polygons (Hafner and Bühler, 2019, 2021), relied on optical SPOT 6/7 satellite imagery. Furthermore,

there have been investigations with external data into the reliability and completeness of mappings from SPOT 6/7 (Hafner et al., 2021). Consequently, an automation of the manual mapping from this imagery would allow for a fast comprehensive documentation of future avalanche periods with background knowledge about how well it works and how much avalanche area approximately is missed. Without an automation it is not feasible to cover large regions quickly. With manual image interpretation (Hafner et al., 2021) it took approximately one hour to manually delineate avalanches in SPOT images covering

a region of $\approx 27.5$ km$^2$. Thus, in this work we develop, describe and apply a deep learning approach for avalanche mapping based on the SPOT 6/7 sensor with the goal to automate the mapping process, so as to cover large areas and eventually operate at country-scale. We developed a variant of DeepLabV3+ (Chen et al., 2018) that takes as input SPOT 6/7 images and a digital





elevation model (DEM), and outputs spatially explicit raster maps of avalanches. For our DeepLabV3+ variant we made the encoder and decoder deformable (Dai et al., 2017), thereby our convolutional kernels adapt according to the underlying terrain,

which is essential in the study of avalanches. In addition to a careful description of the network architecture we evaluate results, compare to previous work, examine the reproducibility of the manually mapped avalanches, and discuss the potential and limitations of our method.

## 2   Data

For training and validating our proposed mapping system we utilize SPOT 6/7 images acquired on 24 January 2018 (referred

to as 2018 in the remainder of this paper, Hafner and Bühler, 2019) and 16 January 2019 (referred to as 2019 from now on, Hafner and Bühler, 2021), together with a set of 24'776 avalanche annotations delineated by manual photo-interpretation. In both cases the images were acquired after periods with very high avalanche danger, i.e., the maximum level 5 of the Swiss avalanche warning system (WSL Institute for Snow and Avalanche Research SLF (ed.), 2021). SPOT 6/7 images have a ground sampling distance (GSD) of 1.5 m and provide spectral intensities in the red, green, blue, and near-infrared (R, G, B, NIR)

wavelengths, at a radiometric resolution of 12 bits. The dataset covers an area of $\approx 12'500$ km$^2$ in 2018 and $\approx 9'500$ km$^2$ in 2019. These two areas partly overlap. Snow reflectance and atmospheric influences exhibit little variability between the two years, but they differ in terms of snow conditions: in 2019 the snow line was at a lower altitude, and consequently there was more dry snow, hardly any wet snow, and fewer glide snow avalanches. As additional input information we use the Swiss national DEM *swissALTI3D*. To match the resolution of SPOT imagery, we resample the DEM (original GSD 2 m) to 1.5 m,

aligned with SPOT 6/7. Its nominal vertical accuracy is 0.5 m below the treeline ($\sim$2100 m a.s.l.) and 1–3 m above the treeline (swisstopo, 2018). We did not apply atmospheric corrections as the water content of the atmosphere is typically low and atmospheric effects therefor relatively minor in winter (Nolin, 2010).

For each mapped avalanche polygon the expert also recorded a score of how well the avalanche was visible, splitting the annotations in three groups: *complete, well visible* outline; *mostly well visible* outline; and *not completely visible* outline,

where significant parts had to be inferred with the help of domain knowledge (see also Bühler et al., 2019). The methodology for manual avalanche mapping follows Bühler et al. (2019) and is described in more detail there. Furthermore, we validated a subset of the initial mapping with independent ground- and helicoper-based photographs as reference (Hafner et al., 2021). We found that for manual mapping based on SPOT images the probability of detection (POD; the probability of a true avalanche being annotated) is 74% for avalanches larger than size 1 (avalanche size is categorised on a scale from 1 to 5, with size 5 the

largest and most destructive ones; for more details see WSL Institute for Snow and Avalanche Research SLF (ed.), 2021). The positive predictive value (PPV; probability of an annotated avalanche having a true counterpart) was 88%, indicating only few false positive annotations (again for size $\geq$2).



## 3 Method

Many overlapping avalanches exist in the dataset whose boundaries cannot be precisely distinguished from each other even by experts. We thus restrict ourselves to identifying all pixels where avalanches have occurred, but do not attempt to group them into individual avalanche events. In terms of image analysis this corresponds to a semantic segmentation task, where each pixel is assigned a class label: *avalanche* or *background* according to its class score. Several deep learning models have been developed for solving such problems and have achieved excellent results in various domains, such as U-Net (Ronneberger et al., 2015), HRNetV2 (Sun et al., 2019) and DeepLabV3+ (Chen et al., 2018).

### 3.1 Model Architecture

On their way downwards, avalanches are constrained and guided by the local terrain. In order to accurately map avalanches from the input data, we therefore propose a deep learning architecture that adapts to the underlying terrain model. We build on state-of-the-art model DeepLabV3+ designed for semantic segmentation and add deformable convolutions that adapt their receptive field size according the input data, i.e. the terrain model in our case.

**DeepLabV3+** is a popular, fully convolutional semantic segmentation model that has been used successfully with a variety of datasets. It features a dilated ResNet encoder as a backbone for feature extraction, in combination with Atrous Spatial Pyramid Pooling module (ASPP). To achieve a wide receptive field able to capture multi-scale context, ASPP employs dilated convolutions at different rates. Before being fed into the decoder, the resulting features are concatenated and merged using a $1 \times 1$ convolution. These high-level features are then decoded, upsampled and combined with high-resolution, low-level features from the first encoder layer. For further details about DeepLabV3+, see Chen et al. (2018).

Our adaptions to the standard DeepLabV3+ include: *deformable kernels* (Dai et al., 2017) in the encoder and decoder as well as a small network with offsets that estimates the appropriate kernel deformations in a data-driven manner, and modifies the decoder such that it can process features from *all backbone layers* (Figure 1). These changes add a modest 1.9M network weights to the 22.4M weights of the standard DeepLabV3+.

The reasoning behind *deformable convolution kernels in the backbone* (Figure 3) is to adapt their receptive fields to the underlying terrain. To obtain deformable convolutions, we introduce an additional 18-channel tensor that encodes the 2D offset of each kernel element at each location i.e., it enables free-form deformations of the kernel, beyond dilation or rotation. The offsets are not fixed a-priori, but calculated as a learned function of the DEM, separately for each feature resolution, by a small additional network branch. By replacing the first convolution in each residual block with a deformable one, we are able to explicitly include the terrain shape encoded in the DEM, but without the need to modify other parts of the architecture, so as to benefit from the pretrained weights of the encoder.

The *augmented decoder* helps our DeepLabV3+ to propagate features along specific directions, in our case this is the possible downhill flow direction of avalanches which can be extracted from the DEM. Hence, we alter the ASPP such that it aggregates features from *all* backbone layers, and increase the receptive field. The new module, which we call *Deformable Spatial Pyramid Flow* (DSPF, Figure 4), performs deformable convolutions at different dilation rates. The deformations are



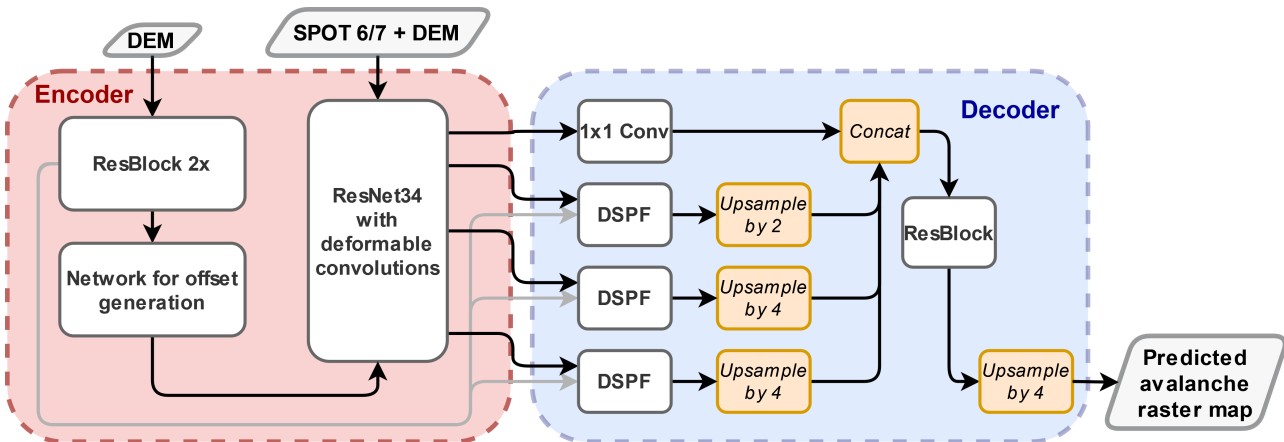

**Figure 1.** Overview of our DeepLabV3+ variant. The encoder is shown in more detail in Figure 3 and the Deformable Spatial Pyramid Flow (DSPF) in Figure 4.

.

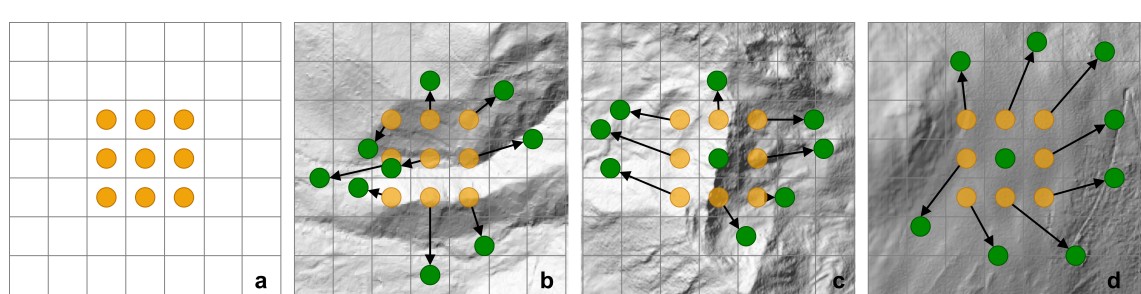

**Figure 2.** For the deformable convolutions, a standard kernel (like the $3 \times 3$ as shown in a) will be adapted according to 2D offsets learned from the underlying DEM. The green dots in b, c and d exemplarily show possible final positions of the kernel elements, the displacement from the standard kernel is illustrated by the black arrows.

.

again obtained from our small network with offsets, based on the DEM. In order to propagate information along the gradient field, we also model the flow direction of an avalanche in the DSFP module of the decoder.

### 3.1.1 Sampling and Data Split

Satellite images are normally too large to be processed with deep learning methods all at once and are therefore cropped to
125 patches of smaller size. Furthermore, a frequent challenge in supervised machine learning approaches is the class imbalance. In our dataset it is very imbalanced: avalanches cover only one 1785th of the entire area covered by SPOT 6/7 imagery. Re-balancing of class frequencies is necessary to make sure our model adequately captures the variability of the avalanche class. We use the following pragmatic strategy to ensure a training set that includes relevant examples, and with sufficient representation





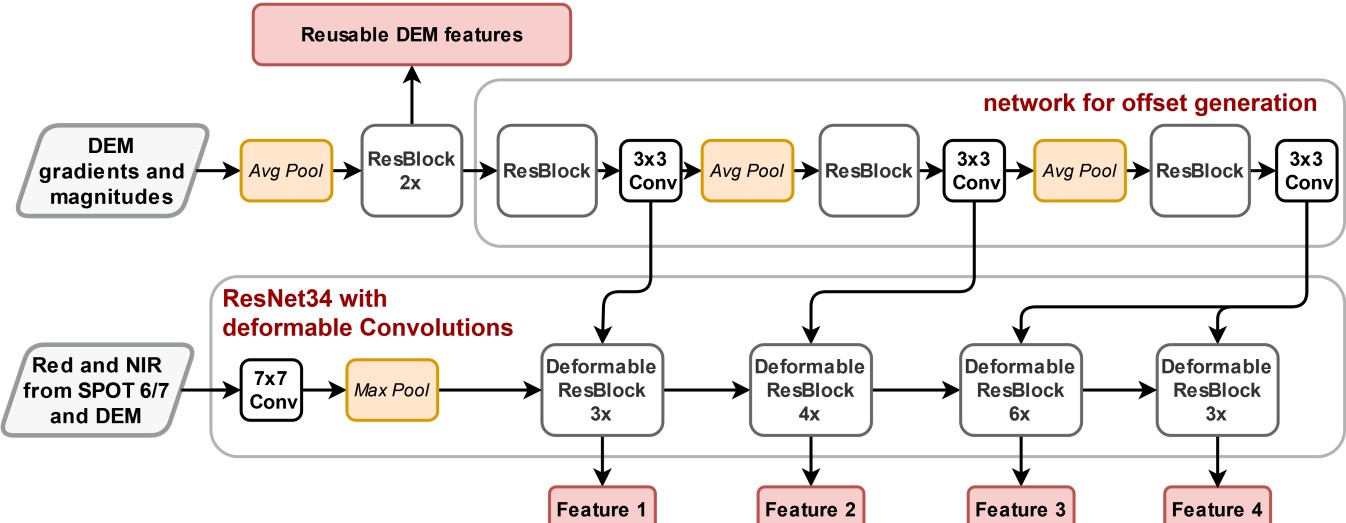

**Figure 3.** Encoder of our DeepLabV3+ in detail.

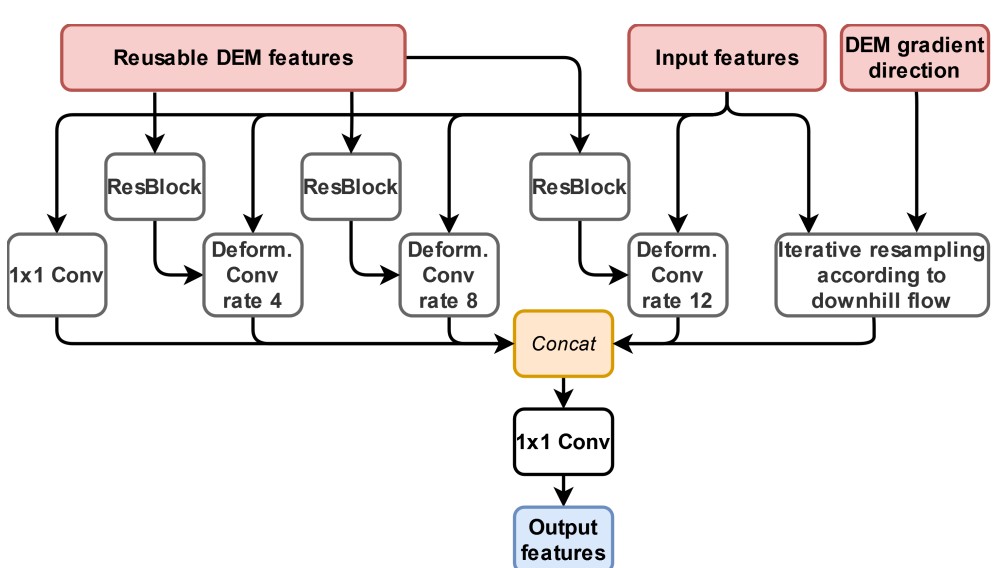

**Figure 4.** Detailed architecture of the Deformable Spatial Pyramid Flow (DSPF) used in the Decoder of our DeepLabV3+ variant.

of both classes: First, we iteratively sample patch centers inside avalanche polygons, while avoiding overlapping patches. In this way, we obtain a set of samples that is not overly imbalanced, with ≈3.5× more background pixels than avalanche pixels. These patches form 95% of our training set. The remaining 5% are sampled randomly in areas without avalanches, to ensure





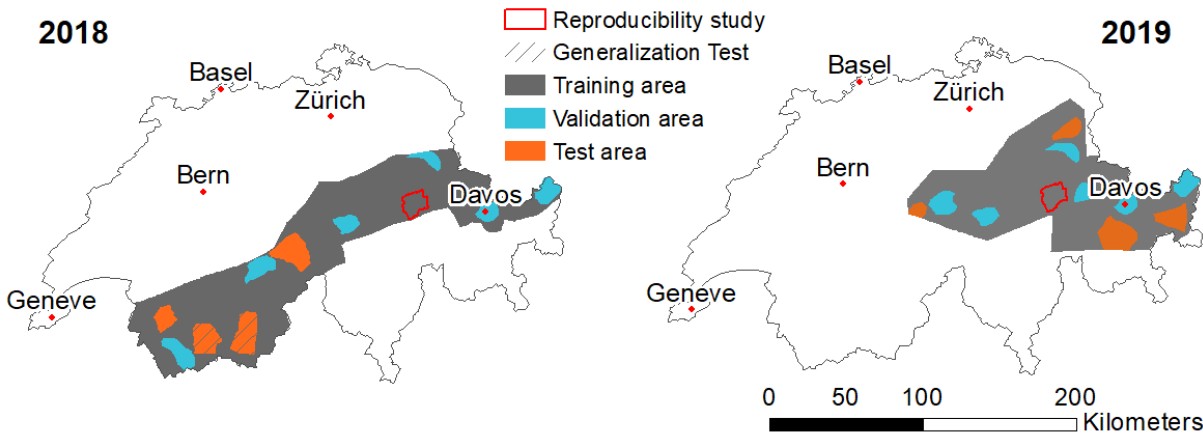

**Figure 5.** Visualization of the disjoint regions for training, validation and testing for both 2018 and 2019. Also shown are: the test region for the generalization experiments, where we had additional data from 6 January 2018, and the regions used to study reproducibility of manual avalanche maps.

also patches without avalanche pixels are seen during training. This leads to an effective ratio of 1:4 between avalanche and background pixels in the 5185 $512 \times 512$ patches of the training set.

As the edges of the patches lack context, they were also given smaller weights when calculating the loss function during training starting 100 pixels from the edge, decreasing the weight linearly to 10% of the base weight given above at the very edge. For our DeepLabV3+ we additionally used deep supervision as in Simonyan and Zisserman (2015), to help the model converge. To increase the model's performance, gradients are accumulated over two iterations before weights are updated. Thereby an effective batch size of 16 (8+8) is reached and the $512 \times 512$ pixel patches may be used (see also section 4.2). Predictions are made for an area specified by a shapefile. To reduce artifacts at the edges of patches, the samples for the predictions overlap by 100 pixels before being cropped.

### 3.1.2 Training

For training and quantitative evaluation, the data were split into mutually exclusive, geographically disjoint regions for training (80%), validation and hyper-parameter tuning (10%) and testing (10%), as depicted in Figure 5. The test set is located completely in regions acquired either only in 2018 or only in 2019, but not in the overlap between the two acquisitions, to prevent memorization (especially of the identical topography).

The network is trained by minimizing a weighted binary cross entropy (BCE) loss (see also 3.1.2), using the Adam optimizer (Kingma and Ba, 2017) for 20 epochs. The base learning rate was initialised to $1 \times 10^{-4}$ and reduced by a factor of 4 after 10 epochs. A summary of the hyper-parameter settings is given in Table 1.





**Table 1.** Summary of training parameters

| Parameter | Value |
|---|---|
| Loss function | Weighted BCE |
| Optimizer | Adam |
| Initial learning rate | $1 \times 10^{-4}$ |
| Effective batch size | 16 |
| Patch size | 512×512 |
| Epochs | 20 |

As a preprocessing step, the input images are normalized channel-wise using the mean and variance values of the entire

150   dataset. Additionally, we flattened the peak in the image histograms caused by the shadow pixels by transforming negative

values $v \rightarrow (-3 \cdot v^2)$, while keeping positive values unchanged.

Even though our training dataset is large, it covers only two avalanche periods and cannot be expected to account for the

whole variety of possible conditions. In order to increase the robustness of the network, we further expand the training set

with synthetic data augmentation. We used randomized rotation and flipping for greater topographic variety, mean-shifting and

155   variance-scaling to simulate varying atmosphere and lighting conditions, as well as patch shifting to increase robustness when

only part of an avalanche is visible. To speed up data loading we used batch augmentation (Hoffer et al., 2019), where the same

sample is read only once and used multiple times with different augmentations computed on the fly.

As mentioned in Section 2 the avalanche polygons come with labels that quantify their visibility in the SPOT data. These

labels are used to re-weight their contributions to the BCE loss as follows: pixels on *complete, well visible* avalanches have

160   weight 2, *mostly well visible* avalanches as well as *background* pixels not on an avalanche have weight 1, and *not completely

visible* avalanches have weight 0.5.

## 4   Results and Discussion

To assess the detection performance of the network, we calculated positive predictive value (PPV, also called precision) and

probability of detection (POD, also called recall) on a pixel level as well as the F1-score. PPV and POD are both based on a

165   standard $2 \times 2$ confusion matrix (Trevethan, 2017). As per pixel metrics take as input a binary mask (avalanche yes or no) and

the network yields scores, we thresholded the predictions at 0.5 before calculating statistics and computed the F1-score as

$$\text{F1} = 2 \cdot \frac{\text{PPV} \cdot \text{POD}}{\text{PPV} + \text{POD}}, \qquad (1)$$

where POD and PPV are defined as

$$\text{POD} = \frac{\text{TP}}{\text{TP} + \text{FN}} \qquad \text{and} \qquad \text{PPV} = \frac{\text{TP}}{\text{TP} + \text{FP}}, \qquad (2)$$

170   where TP is true positive, FP is false positive and FN is false negative.





In this paper the presented pixel-wise metrics (POD, PPV and F1-score) represent the average score over all the patches we tested on. As our dataset is imbalanced and the F1 score non-symmetric, we calculated those metrics for both avalanches and the background. Additionally, we wanted to estimate how many avalanches were detected by each model. Consequently, for the object-based metrics we tested two different measures: we counted an avalanche as detected if 50% or 80% of all pixels within an avalanche from the manual mapping had a score of 0.5 or higher.

## 4.1 Results and generalization ability

Table 2 demonstrates that a standard DeepLabV3+ has a lower POD (0.587) than our method (0.610) while having the same PPV, both for avalanches. This results in an F1-score of 0.612 for the standard DeepLabV3+ and 0.625 for our version with the F1-score for the background being almost identical for both models. For background, the pattern is similar, the POD is slightly better for our method (0.894), compared to the standard DeepLabV3+ (0.888), while the PPV is slightly higher for the standard model (0.900 vs. 0.894). The F1-score is again very similar, as it only differs by one in the third decimal place between our and the standard DeepLabV3+.

For any supervised classification and deep learning methods in particular, the ability to generalize well to new datasets and regions not seen during the training phase is key. To evaluate this, we test our trained model using unseen SPOT 6 imagery of the Mattertal, Val d'Hérens and Val d'Herémence in Valais, Switzerland from 6 January 2018 covering ≈660 km² (see Figure 6). The data were acquired for test purposes after a period with high and very high avalanche danger and the avalanches used for validation have been manually mapped with the same methodology as the others used in this work and described in Bühler et al. (2019). The geographical region with additional data overlaps with data acquired on 24 January 2018, but served as test area before and did not go into training or validation (see Figure 5). The images suffer from distortion in steep terrain as they were part of a usability study for avalanche mapping from optical data (for details see Bühler et al., 2019) and orthorectified by the satellite providers using the height information from the Shuttle Radar Topography Mission (SRTM, OpenTopography, 2013).

The test metrics for predictions on the data from 6 January 2018 were calculated with the standard DeepLabV3+ and the adapted DeepLabV3+. As Table 2 shows, our version generalizes very well, the metrics only differ from tests on the initial dataset in the fourth decimal place. The standard DeepLabV3+ on the other hand, does not generalize so well as the POD and the detection rates per avalanche are lower than for testing on the initial data (see Table 2).

We also investigated object-based metrics for all model variations, when detection means 50% of the avalanche area the models rightly capture between roughly 58% and 69% of all avalanches and between 38% and 51% when detection requires 80% of the area (Table 3). Again the the standard DeepLabV3+ performs slightly worse than our adapted DeepLabV3+, especially when run on data from a new avalanche period (6 January 2018). Therefore, our DeeplabV3+ shows better ability to generalize to new and previously unseen data. The best performance is achieved when considering sunlit avalanche parts only, for both training and testing.

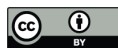



**Table 2.** Segmentation results for the test areas for the standard DeepLabV3+ and our DeepLabV3+, results of predicting on data from a previously unseen avalanche period and variations to our model for the ablation. The metrics shown are the averages from all tested patches.

| Model | SPOT data | training data | patch size | bands | loss function | POD | PPV | F1 | POD back | PPV back | F1 back |
|---|---|---|---|---|---|---|---|---|---|---|---|
| **standard DeepLabV3+** | 24.01.2018 / 16.01.2019 | whole avalanches | 512×512 | R, NIR, DEM | weighted BCE | 0.587 | 0.667 | 0.612 | 0.888 | 0.900 | 0.885 |
| adapted DeepLabV3+ | 24.01.2018 / 16.01.2019 | whole avalanches | 512×512 | R, NIR, DEM | weighted BCE | 0.610 | 0.668 | 0.625 | 0.894 | 0.894 | 0.884 |
| **standard DeepLabV3+** | **06.01.2018** / 16.01.2019 | whole avalanches | 512×512 | R, NIR, DEM | weighted BCE | 0.547 | 0.667 | 0.591 | 0.876 | 0.916 | 0.887 |
| adapted DeepLabV3+ | **06.01.2018** / 16.01.2019 | whole avalanches | 512×512 | R, NIR, DEM | weighted BCE | 0.610 | 0.668 | 0.625 | 0.894 | 0.895 | 0.884 |
| adapted DeepLabV3+ | 24.01.2018 / 16.01.2019 | whole avalanches | **256×256** | R, NIR, DEM | weighted BCE | 0.723 | 0.587 | 0.645 | 0.720 | 0.796 | 0.720 |
| adapted DeepLabV3+ | 24.01.2018 / 16.01.2019 | whole avalanches | **128×128** | R, NIR, DEM | weighted BCE | 0.898 | 0.659 | 0.829 | 0.340 | 0.551 | 0.452 |
| adapted DeepLabV3+ | 24.01.2018 / 16.01.2019 | whole avalanches | 512×512 | R, NIR, DEM | **unweighted BCE** | 0.688 | 0.575 | 0.622 | 0.887 | 0.908 | 0.888 |
| adapted DeepLabV3+ | 24.01.2018 / 16.01.2019 | whole avalanches | 512×512 | **R, G, B, NIR, DEM** | weighted BCE | 0.559 | 0.682 | 0.613 | 0.883 | 0.913 | 0.889 |
| adapted DeepLabV3+ | 24.01.2018 / 16.01.2019 | whole avalanches | 512×512 | **R, G, B, NIR, DEM, Wallis-filtered** | weighted BCE | 0.610 | 0.668 | 0.597 | 0.876 | 0.906 | 0.880 |
| adapted DeepLabV3+ | 24.01.2018 / 16.01.2019 | **release area only** | 512×512 | R, NIR, DEM | weighted BCE | 0.053 | 0.665 | 0.183 | 0.777 | 0.992 | 0.856 |
| adapted DeepLabV3+ | 24.01.2018 / 16.01.2019 | **deposits only** | 512×512 | R, NIR, DEM | weighted BCE | 0.196 | 0.788 | 0.347 | 0.797 | 0.986 | 0.868 |
| adapted DeepLabV3+ | 24.01.2018 / 16.01.2019 | **sunlit avalanche parts only** | 512×512 | R, NIR, DEM | weighted BCE | 0.668 | 0.653 | 0.639 | 0.918 | 0.910 | 0.907 |





**Table 3.** Object-based metrics for selected model configurations.

| Model | SPOT data | Training data | Detection rate 50% of avalanche area | Detection rate 80% of avalanche area |
|---|---|---|---|---|
| standard DeepLabV3+ | 24.01.2018 16.01.2019 | whole avalanches | 0.63 | 0.45 |
| adapted DeepLabV3+ | 24.01.2018 16.01.2019 | whole avalanches | 0.66 | 0.46 |
| standard DeepLabV3+ | 06.01.2018 | whole avalanches | 0.58 | 0.38 |
| adapted DeepLabV3+ | 06.01.2018 | whole avalanches | 0.66 | 0.46 |
| adapted DeepLabV3+ | 24.01.2018 16.01.2019 | sunlit avalanches only | 0.69 | 0.51 |

## 4.2 Ablation studies

To understand how our changes to the standard DeepLabV3+ affect performance we varied the model in different ways and
trained, tested and compared the performance. These results can be found in Table 2. First, we investigated the influence of
the deformable backbone and discovered that including it outperforms the non-deformable backbone configurations of the
standard DeepLabV3+. This is the case in our test areas for 2018 and 2019, but also for testing on the avalanche period from 6
January 2018. Secondly, the avalanches in our network have been weighted (see 3.1.2) according to the quality index assigned
by the manual mapper. To quantify the effects of using weights we ran training with unweighted BCE and observed a decrease
in POD, a slight increase in PPV and overall a smaller F1-score. Additionally, in our adapted version of DeepLabV3+ we
only considered the Red and Near-Infrared from SPOT as well as the DEM as input channels. We cannot test the adapted
DeepLabV3+ without the DEM, as it is explicitly included as an integral part of the network. We analyzed however, how
including all SPOT channels (additionally Blue and Green) and also adding another Wallis filtered channel (to bring out details
in the shade) affects network performance (see Table 2). For our model we found that including more channels did not improve
the performance, rather training time was longer and metrics worse than with the initial channels.

We hypothesize that the proportion of potential avalanche area and context visible in the patches strongly influences network
output. To investigate this, we have trained our model with varying patch size: $512 \times 512$, $256 \times 256$ and $128 \times 128$ pixels
(corresponding to $768 \times 768$, $384 \times 384$ and $192 \times 192$ meters). Quantitative results in Table 2 show the largest patch size
performs best considering metrics for both avalanches and background. When comparing them visually (Figure 7) this is
further supported, as the predictions on the smallest size are patchy, dispersed over the image showing the model is unsure

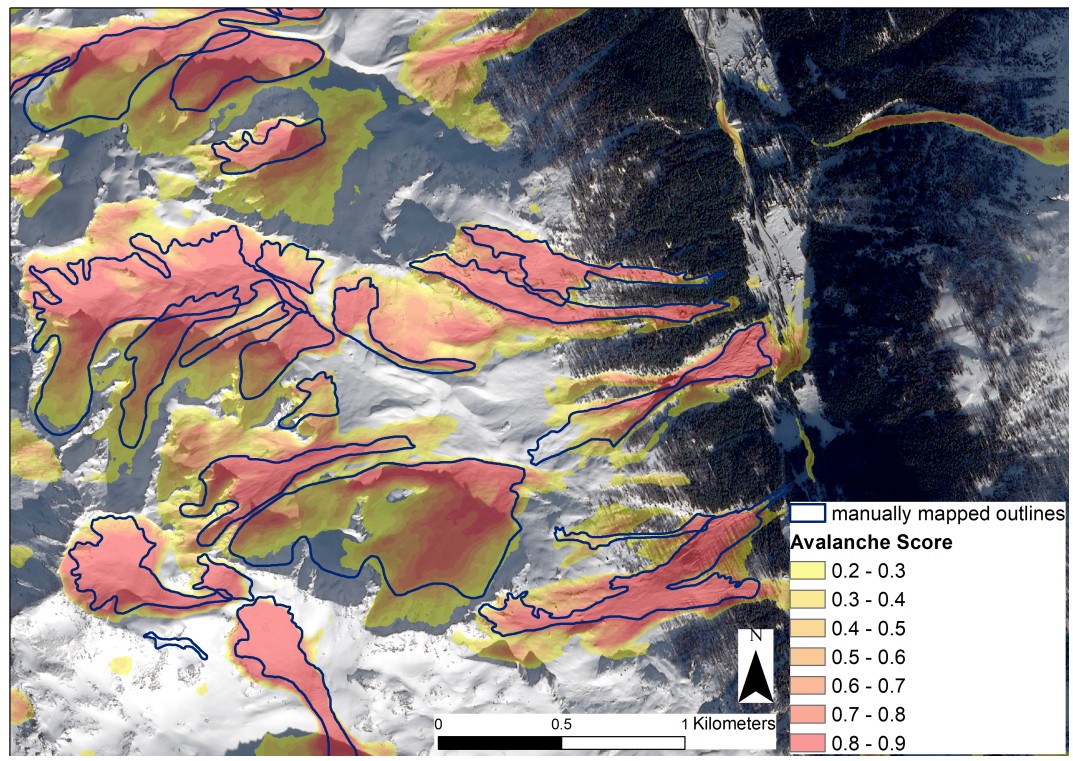

**Figure 6.** An example for the scores when predicting on data from a previously unseen avalanche period from 6 January 2018 (SPOT 6 data © Airbus DS 2018). The darker hues indicate places where the model is more confident. In the illuminated regions those areas almost always correspond to manually mapped avalanches.

.

about the occurrence of avalanches. With increasing context through a larger patch size though, the model becomes more confident and the avalanche borders are distinctly visible.

Subsequently, in order to understand what is better for training the network, we trained on avalanche deposits or release areas only. As deposit area, we assumed the lower third (based on elevation) of each manually mapped avalanche, ignoring those avalanches were the deposit had been inferred. For the release areas, we used the zones identified by Bühler et al. (2019), again disregarding those avalanches where the release zone had been inferred and were therefore uncertain. As results in Table 2 show, performance for predicting all avalanches is a lot worse in both cases. We also observe that PPV and POD are significantly higher when the network is trained on deposits only, rather than trained only on release areas. This resulted in an increase of 0.146 in F1-score and suggests that the original model might also be learning more from texture rich avalanche deposits than from release zones.

Finally, the experts manually mapping the avalanches generally perceived those in the sun as better visible. Hafner et al. (2021) confirmed that and found the POD to be higher roughly by a factor five for avalanches in fully illuminated terrain

**Figure 7.** Comparison of results for four patches when training the network with different patch sizes. The tiles depict (a) the SPOT 6 image, (b) the manually mapped annotations used as reference, (c) the predictions thresholded at 0.5, and (d) the predicted avalanche probability (SPOT 6 data © Airbus DS 2018). Visual inspections show, the model is a lot more confident the larger the patch size.

.

compared to those, at the time of image acquisition, in fully shaded terrain. In order to investigate this further, we used a

Support Vector Machine (SVM) classifier to calculate a shadow mask for both 2018 and 2019. The mask also includes most forested areas due to their speckled sun/shade pattern. Subsequently, we excluded the avalanche parts being located in the shade and trained only with the remaining areas (about one fourth of the avalanche area per year). Calculating the metrics considering only avalanches in illuminated areas, we found an increase of 0.058 in POD, a slight decrease of 0.015 in PPV and





consequently an increase in F1-score of 0.014. The object-based metrics (Table 3) are also slightly better when only considering
sunlit regions.

## 4.3 Reproducibility of manually mapped avalanches

To assess the degree of label noise in our dataset, we conducted a reproducibility experiment on the manually mapped
avalanches to understand how similar the assessment of a given area by different experts would be. In other fields several
comprehensive studies have already been conducted to investigate inter-observer variability, for example for contouring organs
in medical images (Fiorino et al., 1998) or for manual glacier outline identification (Paul et al., 2013). For our investigation
five people attempted to replicate the manual mapping with the same methodology as used before and described in detail in
Bühler et al. (2019). All five mapping experts are very familiar with satellite imagery and/or avalanches and received the same
standardized introductions. The experiment was conducted twice in an area of 90 km$^2$ around Flims, Switzerland, on the 2018
and 2019 SPOT 6/7 imagery (see Figure 5). The area contains avalanches in the shade and in illuminated terrain as well as
all outline quality classes in the initial mappings (Hafner and Bühler, 2019, 2021). The mapping experts did not see another
mapping before having finished theirs.

**Table 4.** F1-scores for the reproducibility investigation: the bold values in the upper right part of the table represent the scores comparing
two expert mappings in illuminated terrain, the lower left values the scores in shaded terrain.

|  | Expert 1 | Expert 2 | Expert 3 | Expert 4 | Expert 5 |
|---|---|---|---|---|---|
| Expert 1 |  | **0.758** | **0.623** | **0.617** | **0.653** |
| Expert 2 | 0.401 |  | **0.711** | **0.723** | **0.724** |
| Expert 3 | 0.232 | 0.198 |  | **0.656** | **0.782** |
| Expert 4 | 0.188 | 0.236 | 0.205 |  | **0.786** |
| Expert 5 | 0.123 | 0.155 | 0.204 | 0.244 |  |

Calculating F1-score (see Formula 1), between all experiment mappings, we found an overall F1-score of 0.381 in illu-
minated and 0.018 in shaded areas. Comparing two expert mappings at a time, the values range from 0.617 to 0.786 in the
illuminated regions and from 0.123 to 0.401 in the shaded regions of our study area (Table 4). The F1-scores of the expert
manual mappings with the initial mapping are in the same range (not shown). The results from 2018 (Figure 8) illustrate that
for some selected avalanches the agreement is very good while, especially in the shade, there is little agreement among experts
on the presence of avalanches.

Reexamining the results from the network now in the light of this experiment, the adapted DeepLabV3+ is equally good
as the experts in identifying avalanches. In other words, we cannot expect a computer algorithm to provide better scores than
the average F1-score of two mapping experts. Even for the avalanches with the highest agreement, a specific boundary line
will usually not match exactly. This makes it hard for any network to learn the localisation of boundaries. We do not yet know



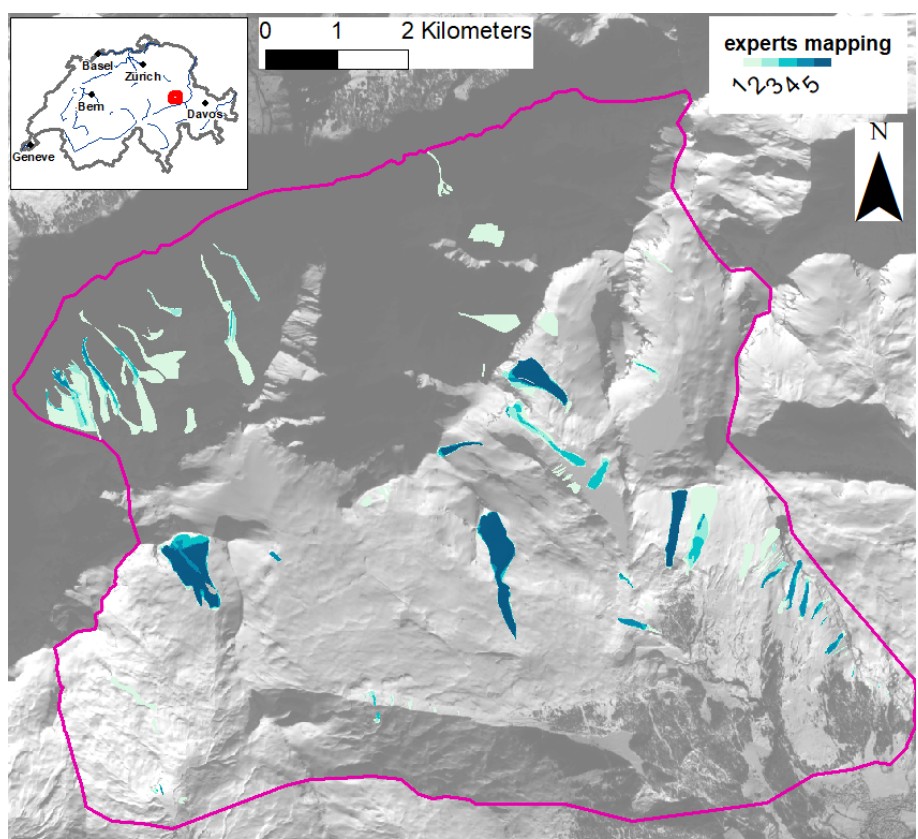

**Figure 8.** Heat map examplarily illustrating expert agreement on avalanche area for avalanches mapped from SPOT in January 2018 (24 January 2018, SPOT 6 © Airbus DS2018). Agreement in the shade (northern part of the study area) is generally lower than in the sunlit areas to the south. The Figure also shows that agreement is very good only on few selected avalanches (depicted in dark blue). For the more detailed location of the reproducibility study area see Figure 5.

.

what exactly causes the differences in avalanche identification between experts. Therefore we plan on conducting a thorough analysis on imagery with different spatial resolutions in the future. This will help to better understand the inherent mapping
uncertainty of avalanches and may give an indication what performance can be expected if training computational detection algorithms on different optical data.

## 4.4 Limitations of this study

The three avalanche periods for which we have SPOT imagery all occurred in January. Those images are relatively close to the winter solstice and therefore have a high percentage of shaded area. (Hafner et al. (2021) identified for 180 km² around Davos
that the share of shaded area at the SPOT 6/7 image acquisition time ranges from 43% at winter solstice to 7% three months later. Even though 2018 includes wet snow and wet snow avalanches, the snow in January is generally colder and drier than

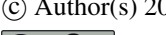



towards spring. Consequently, we do not know how well our model performs under different snow and illumination conditions, for example in spring. Based on the results when training and testing on sunlit avalanche parts only, we see potential for better overall metrics as a smaller portion of the area is shaded. But whether our network already generalizes enough or is biased
towards January conditions and requires retraining with spring SPOT 6/7 we could not yet test.

## 5 Conclusion and outlook

We present a novel deep learning approach for avalanche mapping with deformable convolutions that adapts its notion of the local terrain according to the input digital elevation model (DEM). Experiments at large scale with optical, high spatial resolution (1.5 m) SPOT 6/7 satellite imagery show that our approach achieves good performance (F1-score 0.625) and generalizes
well to new scenes not seen during the training phase (F1-score 0.625). As reference data for training, validating and testing our model we relied on 24'747 manually mapped and annotated avalanches from two avalanche periods on different years. With our adapted DeepLabV3+ we were able to detect 66% of all avalanches. By varying model parameters and the input data we analyzed the impact of different configurations on the mapping result. We found that weighting the avalanches according to the perceived visibility did result in slightly better metrics than when not weighting them. By training on release areas and
deposits only we demonstrated that the network learns more from deposits (F1-score release areas 0.183; F1-score deposits 0.347) and by excluding shaded areas from training we showed that in illuminated terrain both training is easier and test results are better (F1-score 0.639). Furthermore, we investigated expert agreement for manual avalanche mapping in a small reproducibility study and found that agreement on avalanche area is substantially lower than expected. Compared to the model, the agreement between experts is in the same range as the adapted DeepLabV3+ performance.
Our work is an important step towards a fast and comprehensive documentation of avalanche periods from optical satellite imagery. This could substantially complement existing avalanche databases, improving their reliability to perform hazard zoning or the planning of mitigation measures. For the future we aim at conducting a more throughout study investigating expert agreement for manual avalanche identification and its implications for automated avalanche mapping. Additionally, we intend to study the performance of our model on data from different sensors and time periods. Furthermore, we plan on
improving results by masking out areas where avalanche cannot occur using for example modelled avalanche hazard indication data from (Bühler et al., 2022).

*Code and data availability.*  The manually mapped avalanche outlines from 24 January 2018 and 16 January 2019 used by us for training, testing and validation are available on EnviDat (Hafner and Bühler, 2019, 2021). The code used will be published and made available on GitHub with the final publication of this paper.

*Author contributions.*  EH coordinated the study, performed all initial manual mappings, expanded the neural network code originally implemented by PB, and did the statistical analysis. RD, JW and KS advised on the machine learning aspects of the project and critically reviewed



the associated results. RD wrote the script for the Wallis filtering. The reproducibility investigation was initiated by KS, coordinated by EH and both YB and EH were part of its mapping team. EH wrote the manuscript with help from all other authors. EH and YB originally initiated the automation of avalanche mapping from SPOT.

*Competing interests.* The authors declare that they have no conflict of interest.

*Acknowledgements.* We thank Leon Bührle, Benjamin Zweifel as well as Andreas Stoffel for mapping for our reproducibility investigation and Frank Techel for valuable input on its analysis.



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
