# Peer review of "Automated avalanche mapping from SPOT 6/7 satellite imagery with deep learning: results, evaluation, potential and limitations"

_The Cryosphere, 2022_

## Referee Comment (RC3)

[referee-annotated manuscript omitted]

---

## Author Response (AR1)

**Answer to reviewer #1:**

Dear Ron Simenhois,

thank you very much for the comments and suggestions regarding our manuscript. We will in the following discuss and attempt to answer them:

We are aware of the cost of data, the dependency on good acquisition conditions and lower reliability in shaded than illuminated regions for SPOT 6/7 data. Nevertheless, we have chosen to push and forward the use of optical SPOT 6/7 data because we see several major advantages of using optical data over SAR images:

- optical imagery is easier to interpret than SAR imagery.
- in our previous work (Hafner et al., 2021) we have shown that mapping from SPOT6/7 is overall more complete compared to Sentinel-1, which is especially true for avalanches of size 3 and smaller. This is (amongst others) related to the underlying spatial resolution of 1.5m for SPOT 6/7 and approximately 10-15m for Sentinel-1.
- in the same work we investigated which part of an avalanche can typically be identified using Sentinel-1 and found (in accordance with previous studies) that it is mostly the deposit, but may include patches from track and release area. When only using Sentinel-1 data it is therefore neither possible to derive the number of avalanches occurred (several unconnected patches for one avalanche), nor the size of the occurred avalanches (size of patches detected does usually not correspond to avalanche size).
- in conjunction with the above point, unless unambiguous with respect to the terrain the origin and release area of avalanche deposits detected in SAR remain unknown.
- several authors have suspected SAR to be less reliable for detecting dry snow avalanches. Eckerstorfer et al (2022) investigated this in more detail, and they were able to only identify 5.9% of all occurred dry snow avalanches in Sentinel-1 imagery employing manual and automatic detection.
- from optical SPOT data we can therefor get a more complete picture of an avalanche period and with whole outlines give something (more) useful to practice.

We will integrate a summarized version of the above arguments for optical imagery into the revised manuscript.

We have already greatly reduced the details concerning the technical specifications of our model but we will try to make those sections more reader friendly and expand the manuscript with a short non-technical description of a DeepLabV3+.

In the following we will address the specific comments:

Line 78: we refer to the person who manually mapped as expert because the mapping required knowledge on remote sensing, avalanches and the generation of a suitable mapping methodology in order to have comparable results over a large area. The detailed procedure is described in our previous work (Bühler et al., 2019). We will correct this sentence and rephrase it in such a way that what we mean becomes clear to the reader.

Line 84/86: Probabilities may generally also be written in percent with 1 equaling 100%. To avoid confusion, we will still change this passage and use 0.74 instead of 74% in the revised manuscript.

Line 101a: We have added the reference to the ResNet when we first mentioned it in line 42, but we will add it here as well in the revised manuscript.

Line 101b: Resnet is a well-established backbone which is why we went with it. We did test the effect of using a ResNet18, ResNet34 and ResNet50 and found the ResNet34 worked the best. Newer and more sophisticated backbones keep being proposed. Testing multiple other backbones would have been beyond the scope of this paper. For the future this would be an interesting aspect to investigate.

Line 110: the reference to Figure 3 is correct as we want to illustrate where in the network the deformable convolutions are implemented. As Figure 2 explains how deformable convolutions work in detail we will additionally add this reference here in the revised manuscript.

Line 125: We will add a sentence properly introducing the patch size we utilize for our model in the revised version of our manuscript.

Line 127-133: You understood the methodology of our data sampling correctly. We will revise the corresponding section to make it easier for the reader to understand.

Line 129: The "Second" must have gone lost in the editing process, we will correct this in the revised version.

Line 146: We chose to use weighted BCE because weighting the outlines according to their perceived visibility (manual mapping) was our main intention. Given the class imbalance IoU loss would have been a valid choice. We did not try it, as we chose not to extensively focus on varying all model parameters, but agree that for future work it would be interesting to (for example) run the model with IoU loss.  However, as our model metrics are about as good as human experts (section 4.3), we do not expect a huge change in metrics by using a different loss.

Line 150-151: As already mentioned, varying all possible model parameters would be beyond the scope of this paper. Additionally, due to the format we have chosen to not include metrics for all variations that we have played with. In this specific case, the transformation of the shadow pixels increases model performance, though not by a large margin (metrics without transformation: POD: 0.618, PPV: 0.593, F1- score: 0.618).

4.1: We will revisit this section and attempt to bring more clarity in the revised version by reorganizing.

Table 2: The bold fonts signify the model parameters that were varied compared to our "initial" model. We will add this information to the legend in the revised manuscript.

Figure 6: We will replace "avalanche score" with "model confidence" in the revised version of our manuscript

Line 232: We know that the data quality from expert mapping is lower in shaded areas (for more details see comment to section 4.4). As our model cannot know more about avalanches in shaded areas than it has been taught, the model will have lower performance in shaded than in illuminated areas.

Table 4: As the experts mapping did not follow exactly the borders of pixels in the SPOT imagery this is an area-wise comparison of the mapped avalanche polygons. Which in terms of results is comparable to a pixel-wise comparison. To make this clear we will add this information to the revised version of our manuscript.

Figure 8: The color scale of our heat map represents the number of experts who have agreed. We will change the title of the color scale to "number of experts" in the revised version to make this clearer.

4.4 Generally, we have decided to limit the content of this section to the specific limitations of the model and only briefly mention the limitations we have already dealt with in previous work. Concerning the reliability of manual mapping from SPOT6/7 data, we have in Hafner et al. (2021) compared the POD in shaded and illuminated terrain. We know that avalanches are more likely to be missed in the shade (POD: 0.15 shade, 0.86 illuminated, 0.74 overall). We have also mentioned that in line 232. In order to make this clearer to the reader unfamiliar with previous work we will expand this section a bit in the revised manuscript.

Line 285: These numbers are correct and may also be found in Table 2. They are lower than for the whole model as the training data is significantly reduced by using only release areas or only deposits. But even though the metrics are not satisfying with the reduced amount of training data, we were able to show that the model learns more from the deposits. We will make this clearer in the revised version of our manuscript.

**Answer to reviewer #2:**

Dear Edward Bair,

thank you very much for the comments and suggestions regarding our manuscript. We will in the following discuss and attempt to answer the points you raised:

1) We are very well aware of the limitations of optical imagery and their dependence on clear sky during data acquisition. We will clearly point this out in the revised version of our manuscript.

2) Crowns, track and debris can usually be identified in the imagery. Texture, hue and hints of possible damage will probably also play a role. How different experts emphasize those in their search/mapping was not yet investigated.

As our model is trained on the manually mapped avalanches, it tries to identify patterns in this dataset to find avalanches independently later on. What exactly the network is focusing on we do not know, but tests indicate the model seems to be focusing on texture and therefore learning more from deposits than from release areas (see metrics in Table 2). We plan to further investigate how avalanches are recognized and mapped by experts in a follow up study.

3) Our first idea to be able to provide separate metrics for sun and shade has also been using solar geometry. Despite using a DSM with 2m resolution and exact information on image acquisition time, sun azimuth and sun altitude, we found the modeled shade does not represent reality well. As we wanted to avoid distorting our results for areas located at the border of sun and shade, we used SVM instead.

4) We did perform ablation studies using the different SPOT bands and investigated how performance changed with varying input data. We found overall metrics neither improved by additionally including the blue and the green band nor when using only the red, green and blue without the near-infrared band. We therefore concluded, that the near-infrared band possesses important information (in the sun) that by far outweighs the disadvantages (in shaded areas). A more in-depth analysis of this aspect is beyond the scope of our paper and would be interesting to investigate for future work.

In the following we will try to answer the specific comments noted in the provided supplement :

Line 49: for completeness we will add the reference to our study (Bühler et al., 2019), in addition to the provided data citations in the revised manuscript

Line 62/68: as already mentioned in the general comments we will add a short paragraph explicitly describing the limitations relying on optical data, including the inability to acquire data during an event (except for wet snow avalanche periods caused solely by warming during the day).

Line 69: we intended to point out that spectral information is provided in the four bands specified. We will change the sentence to "SPOT 6/7 images have a ground sampling distance (GSD) of 1.5 m and provide information in four spectral bands, namely red, green, blue, and near-infrared (R, G, B, NIR), at a radiometric resolution of 12 bits." in the revised version of our manuscript.

Line 70: In our previous publication (Bühler et al., 2019) we have found the following locational accuracy (sample of 11 GCPs) for the data acquired on 24.01.2018: The achieved accuracy (RMSE) of the GCPs was of 1.23 m in X, 0.83 m in Y and 0.16 m in Z.

Line 72: Yes, we use top of atmosphere reflectance. As you rightly pointed out in the comment to line 76, we use it without correcting for atmospheric effects, because our main focus is texture and the absolute spectral values do not matter for avalanche identification. The description regarding low variability between the years was written with the conditions of the avalanche period (on/ close to the ground) in mind. We did not intend to make a statement about atmospheric conditions that influence the spectral values in the satellite imagery, which could be analyzed with independent data such as AOD. We will correct this information to make this clear to the reader in the revised version of our manuscript.

Line 76: as proposed we will delete this sentence in the revised version

Line 83: we have provided the equations for POD and PPV in section 4 where we introduce them for our work. We will add a reference to those formulas here as well in the revised version of our manuscript.

Line 90: see general answer 2): we will add a sentence elaborating on this in the revised version

Line 113/119: we will correct the language mistakes as proposed in the revised version

Line 125: You are right, patch size is dependent on the spatial resolution as well as computational resources. In order to be more specific and make what we mean clear we will replace this sentence. It will be included in the proper introduction of the patch size, that we have already promised to RC2, in the revised version of our manuscript: "Given the proposed model architecture and the available computational resources, we are able to simultaneously process batches of 2 image patches per GPU of up to 512×512 pixels at training time, which translates into an area of 589'824 $m^2$ at the spatial resolution of SPOT 6/7 images."

Line 127: we are sorry, but we do not know where you think the "/" is missing

References:

Bühler, Y., Hafner, E. D., Zweifel, B., Zesiger, M., and Heisig, H.: Where are the avalanches? Rapid SPOT6 satellite data acquisition to map an extreme avalanche period over the Swiss Alps, The Cryosphere, 13, 3225–3238, https://doi.org/10.5194/tc-13-3225-2019, 2019

Eckerstorfer, M., Oterhals, H., Müller, K., Malnes, E., Grahn, J., Langeland, S., and Velsand, P.: Performance of manual and automatic detection of dry snow avalanches in Sentinel-1 SAR images. Cold Regions Science and Technology. 198. 103549. 10.1016/j.coldregions.2022.103549, 2022.

Hafner, E. D., Techel, F., Leinss, S., and Bühler, Y.: Mapping avalanches with satellites – evaluation of performance and completeness, The Cryosphere, 15, 983–1004, https://doi.org/10.5194/tc-15-983-2021, 2021.